# Genome-wide association study of knee pain identifies associations with *GDF5* and *COL27A1* in UK Biobank

Weihua Meng [1,9], Mark J. Adams [2,9], Colin N.A. Palmer[1], The 23andMe Research Team, Jingchunzi Shi[3], Adam Auton[3], Kathleen A. Ryan[4], Joanne M. Jordan [5], Braxton D. Mitchell[4,6], Rebecca D. Jackson[7], Michelle S. Yau[8], Andrew M. McIntosh [2] & Blair H. Smith [1]

Knee pain is one of the most common musculoskeletal complaints that brings people to medical attention. Approximately 50% of individuals over the age of 50 report an experience of knee pain within the past 12 months. We sought to identify the genetic variants associated with knee pain in 171,516 subjects from the UK Biobank cohort and seek supporting evidence in cohorts from 23andMe, the Osteoarthritis Initiative, and the Johnston County Osteoarthritis Project. We identified two loci that reached genome-wide significance in the UK Biobank: rs143384, located in *GDF5* ($P = 1.32 \times 10^{-12}$), a gene previously implicated in osteoarthritis; and rs2808772, located near *COL27A1* ($P = 1.49 \times 10^{-8}$). These findings were supported in cohorts with self-reported osteoarthritis/radiographic knee osteoarthritis without pain information. In this report on genome-wide association of knee pain, we identified two loci in or near *GDF5* and *COL27A1* that are associated with knee pain.

---

[1] Division of Population Health and Genomics, School of Medicine, University of Dundee, Dundee, UK. [2] Division of Psychiatry, Edinburgh Medical School, University of Edinburgh, Edinburgh, UK. [3] 23andMe, Inc., Mountain View, CA, USA. [4] Department of Medicine, University of Maryland School of Medicine, Baltimore, MD, USA. [5] Department of Medicine, University of North Carolina School of Medicine, Chapel Hill, NC, USA. [6] Geriatric Research, Education and Clinical Center, Veterans Affairs Medical Center, Baltimore, MD, USA. [7] Division of Endocrinology, Diabetes and Metabolism, The Ohio State University, Columbus, OH, USA. [8] Institute for Aging Research, Hebrew SeniorLife, Harvard Medical School, Boston, MA, USA. [9] These author contributed equally: Weihua Meng and Mark J. Adams. A full list of the 23andMe Research Team members appears at the end of the paper. Correspondence and requests for materials should be addressed to W.M. (email: w.meng@dundee.ac.uk)

The knee supports body weight when walking, standing upright and bending. Knee pain describes a specific area of pain inside the knee or diffuse pain around knee area[1]. It is one of the most common musculoskeletal complaints that bring people to medical attention[2]. The knee pain experience varies from person to person and can present as a dull ache to a sharp, stabbing pain and from intermittent weight bearing pain to persistent pain[3].

Knee pain is highly prevalent in older individuals, with ~50% of individuals over the age of 50 reporting an experience of knee pain within the past 12 months[4]. In one US general population cohort, knee pain prevalence has increased from 15.7 to 32.9% in females and from 8.7 to 27.7% in males between 1983 and 2005, regardless of knee osteoarthritis status[5]. In another estimate, the prevalence of knee osteoarthritis in the USA increased from 8% in 1950s to 16% currently[6]. There are over eight million patients suffering from knee osteoarthritis in the UK[7]. According to the Global Burden of Diseases 2016, osteoarthritis including knee osteoarthritis is the twelfth leading cause of years of life lived with disability globally[8]. It is estimated that ~50% of all people with knee osteoarthritis have reported knee pain symptoms and of those without knee osteoarthritis, 20% have reported knee pain[5]. There are many underlying mechanisms that can cause knee pain, including injuries, gout and infection, as well as arthritis. Among these, osteoarthritis is the most common cause, particularly in people over the age of 50[5]. People with knee pain will experience progressive loss of knee function and declining quality of daily life, and display increasing dependence in daily activities[9]. Further, knee pain caused by osteoarthritis frequently accompanies pain in other joints, such as hips and hands, which further reduces quality of life[10]. The disease has generated huge economic burdens to the health care systems across the world. For example, although no figures exist specifically on knee osteoarthritis, the total direct cost of osteoarthritis as a whole in the UK in 2010 was around £1 billion and the corresponding total indirect cost of osteoarthritis in 2010 was over £3.2 billion[11].

Epidemiological studies have suggested multiple risk factors for knee pain, including female sex, age, obesity, previous knee injuries, knee-straining work and smoking[12]. Similar risk factors are reported in studies of knee osteoarthritis specifically, which also included kneeling and squatting as further risk factors[12,13]. With aging populations and increasing rates of obesity, the prevalence of knee pain is likely to increase. Psychological factors are also important risk factors of knee pain[14]. These environmental and lifestyle factors are likely to interact with genetic factors, and are important to understand in genetic association studies.

Genetic studies to date have focused on knee osteoarthritis, but not knee pain more generally. Studies in siblings have reported heritabilities for knee osteoarthritis as high as 0.62[15]. In a recent large twin study, 45% of the respective variation for severe knee osteoarthritis requiring joint replacement could be explained by genetic factors[16]. The genetic architecture of knee osteoarthritis was considered to follow an additive genetic model, involving multiple genes or loci but each with small effect size[17]. Candidate genes, including GDF5, COL9A1, IL1B, IL1RN, LRCH1, CLIP, TNA and BMP2, have been reported to be associated with knee osteoarthritis[18–22]. Genome-wide association studies (GWAS) have also reported that the GDF5, DVWA, HLA-DQB1, BTNL2, COG5, MCF2L, TP63, FTO, SUPT3H/RUNX2, GLN3/GLT8D1 and LSP1P3 genes contribute specifically to knee osteoarthritis[23–29]. Recently, Zengini et al. reported nine novel genetic loci associated with osteoarthritis based on five different osteoarthritis definitions according to the self-reported status questionnaire and the Hospital Episode Statistics data from the UK Biobank cohort[30]. However, these analyses did not specifically focus on the knee area.

To identify the genetic variants associated with knee pain, we conducted a GWAS using the large UK Biobank cohort. We defined knee pain as 'knee pain in the last month interfering with usual activity', based on the information available from the study questionnaire. Since there are no knee pain GWAS summary statistics available, we chose to support our genome-wide findings on knee pain using three independent cohorts that defined osteoarthritis using either questionnaire data (i.e. 23andMe) or radiographic criteria (i.e. the Osteoarthritis Initiative (OAI) and the Johnston County Osteoarthritis Project (JoCo)).

Here, we use GWAS on knee pain to screen for genetic variants; similar approaches have been taken for headache and back pain using the UK Biobank cohort[31,32].

## Results

**GWAS results.** A total of 501,708 UK Biobank participants were invited to respond to the pain questionnaire during the initial assessment visit (2006–2010). Among those who responded, 29,995 participants selected the 'Knee pain' option (cases), and 197,149 participants selected the 'None of the above' option (controls). After removing samples from non-British participants, those who were related with another individual in the cohort and those who failed quality control (QC), we identified 22,204 cases (12,062 males and 10,142 females) and 149,312 controls (71,480 males and 77,832 females) for the GWAS association analysis and there were 15,377,520 single nucleotide polymorphism (SNPs) available for the GWAS analysis. The genomic control value (lambda) was 1.06.

Table 1 summarises the clinical characteristics of these cases and controls. There were statistical differences ($P < 0.001$) in sex, age and body mass index (BMI) between cases and controls in the UK Biobank samples.

We identified two SNP clusters that were associated with knee pain, with genome-wide significance ($P < 5 \times 10^{-8}$, Fig. 1, Table 2). Four independent significantly associated SNPs within two clusters are shown in Table 2. All significantly associated SNPs ($N = 107$) in the discovery stage are shown in Supplementary Data 1.

The most significantly associated SNP cluster was in the GDF5 gene in the chromosome 20q11.22 region with a P value of $1.32 \times 10^{-12}$ for rs143384 (A allele, beta: −0.008). The second most significantly associated cluster was in the LOC105376225 gene (near the COL27A1 gene) in chromosome 9 with a lowest P value of $1.49 \times 10^{-8}$ for rs2808772 (A allele, beta: 0.006). The regional plots for loci in GDF5 and COL27A1 are shown in Figs. 2 and 3.

The Q–Q plot of the GWAS in the discovery stage is shown in Fig. 4. The SNP-based heritability of knee pain was 0.08 (standard error = 0.03).

In the independent cohort 1 (23andMe, self-reported osteoarthritis), the P value for rs143384 was $2.44 \times 10^{-9}$ in the 23andMe

---

**Table 1 Clinical characteristics of knee pain and controls in the UK Biobank**

|  | UK Biobank | | |
|---|---|---|---|
| Covariates | Cases | Controls | P |
| Sex (male: female) | 12,062:10,142 | 71,480:77,832 | <0.001 |
| Age (years) | 58.3 (7.64) | 56.9 (7.97) | <0.001 |
| BMI (kg m⁻²) | 28.6 (4.88) | 26.7 (5.00) | <0.001 |

A χ² test was used to test the difference of gender frequency between cases and controls and an independent t test was used for other covariates. Continuous covariates were presented as mean (standard deviation).
BMI body mass index

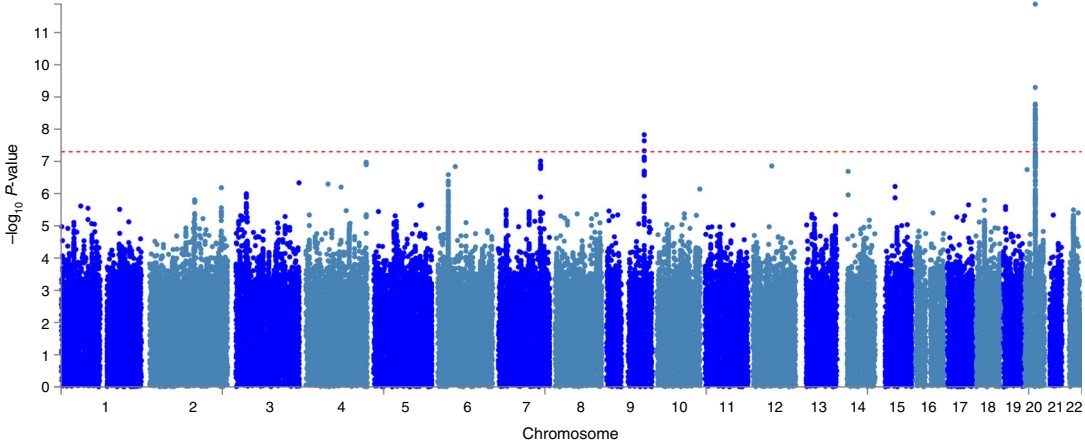

**Fig. 1** Manhattan plot of the GWAS on knee pain using the UK Biobank cohort

| Chromosome | Gene | SNPID | Effective allele in all cohorts | Effective allele frequency discovery cohort | P value (beta) UK Biobank discovery cohort | P value (beta) 23andMe independent cohort 1 | P value (beta) OAI–JoCo independent cohorts 2 |
|---|---|---|---|---|---|---|---|
| 9 | *LOC105376225* (near *COL27A1*) | rs919642 | A | 73.2% | $2.29 \times 10^{-8}$ (0.007) | $4.84 \times 10^{-12}$ (0.028) | 0.88 (−0.008) |
| 9 | *LOC105376225* (near *COL27A1*) | rs2808772 | A | 52.5% | $1.49 \times 10^{-8}$ (0.006) | $4.43 \times 10^{-5}$ (0.014) | 0.36 (0.041) |
| 20 | *GDF5* | rs143384 | A | 60.0% | $1.32 \times 10^{-12}$ (−0.008) | $2.44 \times 10^{-9}$ (−0.021) | 0.01 (−0.12) |
| 20 | *GDF5* | rs6120946 | A | 78.2% | $6.81 \times 10^{-9}$ (−0.008) | 0.00071 (−0.014) | 0.0074 (−0.16) |

**Table 2 Summary of the four independent and significant SNPs associated with knee pain in the *GDF5* and *COL27A1* regions**

OAI–JoCo, the joint Osteoarthritis Initiative–Johnston County Osteoarthritis cohorts

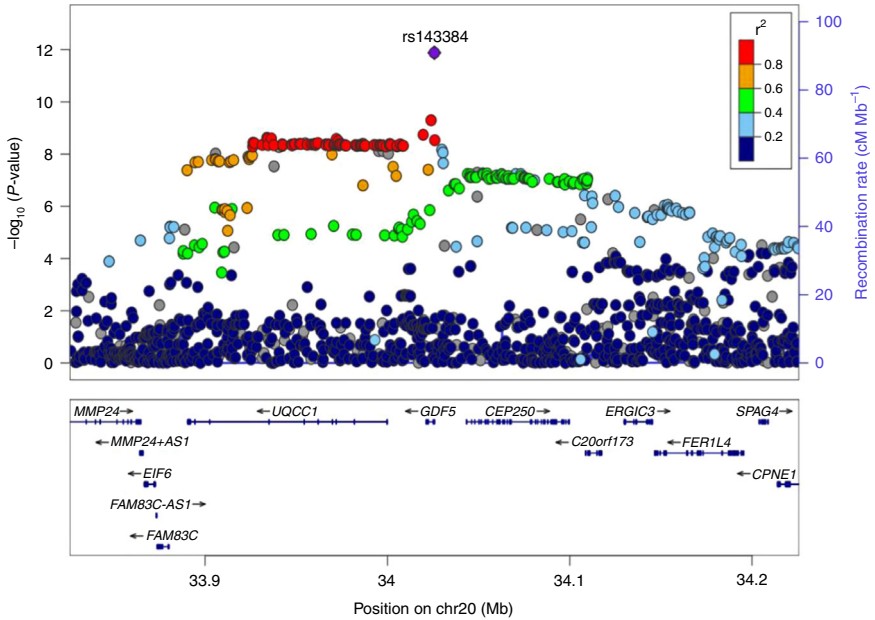

**Fig. 2** Regional plot of the *GDF5* gene region based on the GWAS on knee pain using the UK Biobank cohort

cohort and in the independent cohorts 2 (OAI and JoCo cohorts, radiographic knee osteoarthritis), the *P* value for rs143384 was 0.01 in the combined OAI and JoCo cohorts. The *P* values for rs2808772 were $4.43 \times 10^{-5}$ in the 23andMe (self-reported osteoarthritis) and 0.36 in the combined OAI and JoCo cohorts (radiographic knee osteoarthritis) (Table 2).

**Gene, gene-set and tissue expression analysis by FUMA**. In the gene analysis, all the SNPs that are located within genes were mapped to 19,436 protein coding genes. *GDF5* demonstrated the strongest association, with a *P* value of $1.09 \times 10^{-11}$. The eleven associated genes with *P* values $<3 \times 10^{-6}$ (0.05/19436) were *GDF5, UQCC1, CEP250, PODXL, C20orf173, SPAG4, MTMR3*,

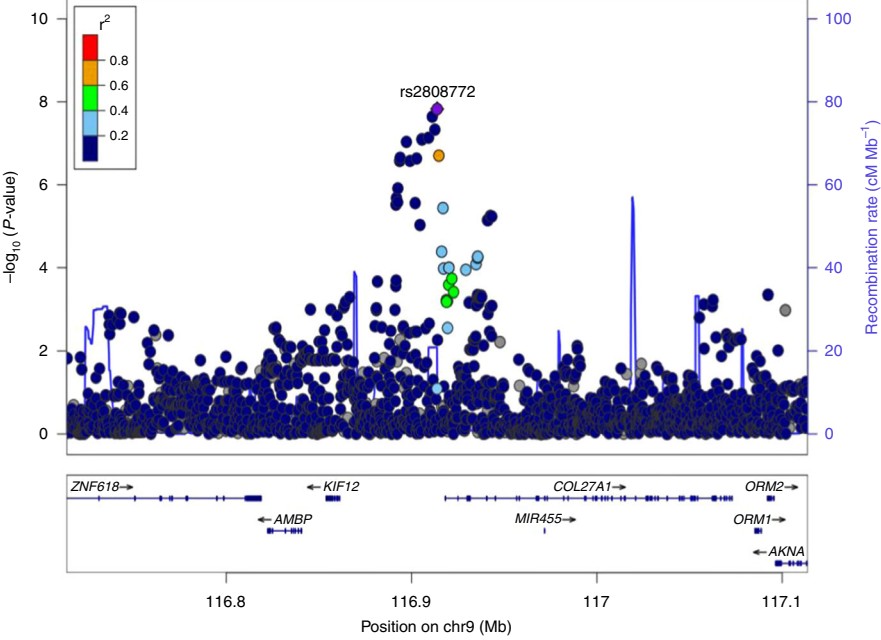

**Fig. 3** Regional plot of the *COL27A1* gene region based on the GWAS on knee pain using the UK Biobank cohort

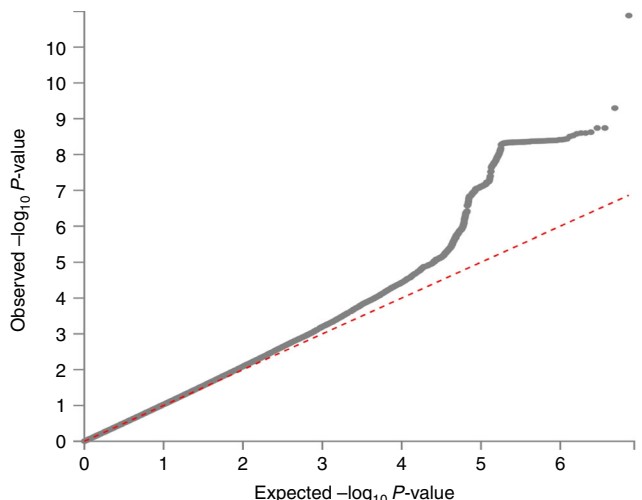

**Fig. 4** Q-Q plot of the GWAS on knee pain using the UK Biobank

*ERGIC3*, *FBLN2*, *CPNE1* and *CDC42SE2*. The results are included in Supplementary Table 1.

In the gene-set analysis, a total of 10,894 gene sets were tested. The regulation pathway of breast_cancer_20q11_amplicon demonstrated a *P* value of $2.59 \times 10^{-8}$ and this was the only gene set with a statistically significant association ($P < 5 \times 10^{-6}$ (0.05/10,894)). The top ten gene sets from this analysis are shown in Supplementary Table 2.

In the tissue expression analysis, none of the tissue types demonstrated statistically significant associations ($P < 0.001$), either in the expression analysis of 30 general tissue types from multiple organs or in the 53 specific tissue types within some of these organs. See Supplementary Figs. 1 and 2.

**Genetic correlation analysis by LD Hub.** We identified multiple significant and negative genetic correlations for knee pain with other traits (Supplementary Data 2). The genetic correlations ($r_g$) surviving multiple testing correction were: Years of schooling

2016 ($r_g = -0.29$, $P = 4.97 \times 10^{-8}$), college completion ($r_g = -0.36$, $P = 6.55 \times 10^{-6}$), age of having first baby ($r_g = -0.30$, $P = 1.92 \times 10^{-5}$).

## Discussion

In the first reported GWAS of knee pain using the UK Biobank resource, we identified variants in or near *GDF5* and *COL27A1*, which were subsequently supported in osteoarthritis cohorts from the 23andMe, OAI and JoCo cohorts. In addition, we found that knee pain was genetically and negatively correlated with a number of socioeconomic factors, such as years of schooling and college completion.

The generic pain question used by the UK Biobank is useful as a screening tool and a useful step to test whether heterogeneous pain phenotypes (such as knee pain) have genetic components at all. The same question has been used to identify the genetic variants of broadly defined headache, and the findings were similar to those for well-defined migraine phenotypes[31,33]. The benefit of using UK Biobank on heterogeneous phenotypes will allow researchers to overcome potential issues with reduced power due to heterogeneity by using very large numbers to cut through the statistical noise.

In this GWAS, we have identified two loci for knee pain. The top locus was in the *GDF5* gene in chromosome 20q11.2 with a lowest *P* value of $1.32 \times 10^{-12}$ for rs143384, while the locus itself was 140 kb long spanning from the *UQCC1* gene to the *GDF5* gene containing 104 genome-wide positive SNPs (Supplementary Data 1). The *GDF5* gene encodes a secreted ligand of the transforming growth factor-beta (TGF-beta) superfamily of proteins[34]. This protein not only regulates the development of numerous tissue and cell types, but also promotes the maintenance and repair of synovial joint tissues, particularly cartilage and bones[34,35]. Mutations in the gene can cause cartilage or bone related disorders, such as chondrodysplasia, acromesomelic dysplasia and brachydactyly, suggesting a protective role in skeletal development[34]. The *GDF5* gene has been repeatedly reported to be associated with osteoarthritis through genetic studies[18,28]. Functional studies have suggested that knee morphology is profoundly affected by Gdf5 absence in mice models, and

downstream regulatory sequences mediate its effects by controlling Gdf5 expression in knee tissues[36]. It was also suggested that osteoarthritis susceptibility mediated by variants in the GDF5 gene was not restricted to cartilage but joint wide[37]. Recently, Capellini et al. combined the transgenic mice model with population genetic analyses in humans to identify a GDF5 enhancer that influences human growth and osteoarthritis risk[38]. Overall, there is sufficient and solid biological evidence relating the GDF5 gene with knee osteoarthritis, and we assume that this finding is due to detection of knee pain caused by osteoarthritis, rather than other pathologies.

The second SNP cluster was in the LOC105376225 area (which is next to the COL27A1 gene) in chromosome 9 with a lowest P value of $1.49 \times 10^{-8}$ for rs2808772. There have been no specific studies published about LOC105376225 and its relationship with knee pain or knee osteoarthritis. However, the neighbouring COL27A1 gene is clearly a good candidate gene. This gene encodes a member of the fibrillar collagen family, and plays a role during the calcification of cartilage and the transition of cartilage to bone[39]. Mutations in the COL27A1 gene have been reported to be associated with the Steel syndrome. This syndrome is characterised by bone changes, such as bilateral hip and radial head dislocations, short stature, characteristic facies, fusion of carpal bones, scoliosis, pes cavus and cervical spine anomalies[40]. Further, the gene was reported to be associated with knee osteoarthritis in the first stage, but did not replicate in the second stage in a recent GWAS study on knee osteoarthritis[29]. Thus, our large study on knee pain has suggested that the COL27A1 gene might play a contributing role for lesions in the knee area. Importantly, polymorphisms in the gene have been associated with tendinopathy around the ankle joint[41]. The concordance between radiographically defined knee osteoarthritis and knee pain is quite poor, with between 15 and 81% of patients diagnosed by radiographic methods having pain symptoms[42]. It is therefore likely that many people reporting knee pain have pain that is not bone or cartilage related, but tendon related. It is possible that variants in the COL27A1 gene could eventually affect the quality of the tendons such as strength or flexibility, which might lead to poor tendon function, or their stability of the attachment to the patella, which might also be knee pain related. A recent study has suggested that a collagen gene COL11A2 play a role in pain sensitisation after the development of osteoarthritis[43]. Interestingly, the COL27A1 gene was not mentioned in GWAS on osteoarthritis using the UK Biobank dataset[44].

Our study focused on knee pain as a broad phenotype and the genes that we identified are suggested to be related to knee osteoarthritis. This suggests that the phenotype we chose was genetically similar to the phenotype of knee osteoarthritis. The relationship between knee pain and knee osteoarthritis deserves further investigation. Studies have shown that people with end-stage knee osteoarthritis all presented with knee pain[45], but this might not be the case for early stage knee osteoarthritis. As described above, 20% of knee pain was not caused by knee osteoarthritis and only 50% of knee osteoarthritis patients with radiographic evidence had knee pain symptoms[5]. In addition, a study has reported that 86% of people reporting knee pain will develop knee osteoarthritis over 12 years[46]. Interestingly, the hospital-diagnosed ($N = 10,083$ cases) osteoarthritis and self-reported ($N = 12,658$ cases) osteoarthritis in the UK Biobank were only half the size of those reported with knee pain in this study ($N = 22,204$)[30]. This indicates the importance of treat knee pain as a single entitle (like back pain) instead of solely being a symptom of knee disorders. The Neale lab (http://www.nealelab.is/uk-biobank) has performed over 2000 traits using the UK Biobank dataset treating knee pain and knee osteoarthritis as individual phenotypes.

The gene analysis by FUMA also supports our finding that GDF5 was the strongest gene for knee pain. The gene-set analysis by FUMA revealed that the regulation pathway of Nikolsky_breast_cancer_20q11_amplicon signalling was associated with the phenotype we use. We noticed that GDF5 and this amplicon were both located in chromosome 20q11 area and it was reported that GDF5 protein regulates TGF-beta dependent angiogenesis in breast carcinoma MCF-7 Cells[47].

The SNP-based heritability for knee pain was 0.08 in our study, which is the first report of its kind. This low heritability suggests the greater role of environmental than genetic factors in development of knee pain, though this method of estimating heritability is likely to miss some of the genetic contribution. We identified that knee pain was genetically and negatively correlated with a number of phenotypes such as years of schooling, college completion and age of having first baby. This means that those with more years of schooling, those with completed college education, and those who were older when they had their first baby were less likely to report current troublesome knee pain. These factors could be related to lifestyle and occupation.

Using the CaTS power calculator (http://csg.sph.umich.edu/abecasis/cats/), we had 80% power to identify SNP associations with a significance level of $5 \times 10^{-8}$, based on 22,204 cases and 149,312 controls, assuming an additive model, a minor disease allele frequency of 0.20, a genotypic relative risk of 1.06 and an estimated prevalence of knee pain in the general population of 0.2.

The major limitation of this study was that we used different (though similar) phenotypes in discovery and supporting stages. This was because of the phenotypic information that was available in the relevant and available cohorts, and the lack of any existing relevant cohorts or datasets examining knee pain as a phenotype. We defined knee pain cases and controls based on the responses by UK Biobank participants to a specific pain question. This question focused on knee pain occurrence, sufficient to cause interference with activities, during the previous month. The question does not ask information of the severity and frequency and the exact area of knee pain. Therefore, our phenotyping should be considered as widely defined. The situation was similar for the 23andMe cohort, in which disease status was also self-reported via survey and self-reported osteoarthritis in the 23andMe cohort was not specific to the knee. Self-reported knee pain has been widely used in other studies as well, though not for studies of genetic associations[48,49]. The OAI and JoCo assessed radiographic evidence of knee osteoarthritis but with limited sample size. Although not true replications, these independent cohorts suggest possible overlapping risk alleles among knee pain, general osteoarthritis and knee osteoarthritis.

In conclusion, we have identified two loci (GDF5 and COL27A1) for knee pain in a GWAS using the UK Biobank resource and found evidence to support them in the 23andMe, OAI and JoCo cohorts. In addition, we found several significant and negative genetic correlations between knee pain and a number of educational phenotypes, suggesting that the genetic aetiology of knee pain may also be related to these traits.

## Methods

**The participants and genetic information of all cohorts**. *Discovery cohort—UK Biobank*: Over 500,000 people aged between 40 and 69 years were recruited by the UK Biobank cohort in 2006–2010 across England, Scotland and Wales. All participants provided informed consent that their health records could be accessed for research purposes. Further information about the UK Biobank cohort can be found at www.ukbiobank.ac.uk. Ethical approval was granted by the National Health Service National Research Ethics Service (reference 11/NW/0382).

DNA extraction and QC were standardised and the detailed methods can be found at http://www.ukbiobank.ac.uk/wp-content/uploads/2014/04/DNA-Extraction-at-UK-Biobank-October-2014.pdf. The Wellcome Trust Centre for

Human Genetics at Oxford University was in charge of standard QC procedures for genotyping results. The detailed QC steps can be found at http://biobank.ctsu.ox.ac.uk/crystal/refer.cgi?id=155580.

In July 2017, the UK Biobank released the genetic information (including directly genotyped genotypes and imputed genotypes) of 501,708 samples to all approved researchers. The detailed QC steps of imputation were described by Bycroft et al.[50].

*Independent cohort 1: 23andMe Inc*: The 23andMe company is a privately held personal genomics and biotechnology company based in the USA. It includes more than 1,500,000 genotyped subjects who have consented to participate in research. All participants included in the analyses provided informed consent and answered surveys online according to 23andMe's human subjects protocol, which was reviewed and approved by Ethical & Independent Review Services, a private institutional review board. The DNA extraction from saliva and the QC of the genotyping and imputation results were all based on the company's standardised procedures. Further methodological details can be found in the supplementary file of a previous publication[51].

*Independent cohorts 2*: (1) The OAI is a prospective longitudinal study designed to identify risk factors for the incidence and progression of symptomatic tibiofemoral knee osteoarthritis. Participants aged between 45 and 79 years were recruited at four different clinical sites in the USA. Details of the study protocol, including recruitment procedures and eligibility criteria are available on the OAI web site. (http://oai.epi-ucsf.org/datarelease/docs/StudyDesignProtocol.pdf). (2) The JoCo is an ongoing, community-based study of the occurrence of knee and hip osteoarthritis in African American and Caucasian residents, aged 45 years and above from Johnston County, North Carolina in the USA. A total of 3068 individuals were recruited at baseline. A detailed description of the cohort has been reported[52]. All participants provided informed consent.

Standard procedures of imputation and QC were applied when genotyping OAI and JoCo samples. The detailed description of the cohorts has been previously published[29].

**Phenotypic information of all cohorts in this study**. *Discovery cohort—UK Biobank*: We used a bespoke pain-related questionnaire adapted by the UK Biobank, which included the question: 'in the last month have you experienced any of the following that interfered with your usual activities?' The options were: (1) Headache; (2) Facial pain; (3) Neck or shoulder pain; (4) Back pain; (5) Stomach or abdominal pain; (6) Hip pain; (7) Knee pain; (8) Pain all over the body; (9) None of the above; (10) Prefer not to say. More than one option could be selected. (UK Biobank Questionnaire Field ID: 6159)

The knee pain cases in this study were those who selected the 'knee pain' option for the above question, regardless of whether they had selected other options.

The controls in this study were those who selected the 'None of the above' option.

Independent cohort 1—23andMe, Inc: The 23andMe cohort used an online survey to determine the phenotypic status of osteoarthritis of all participants.

Cases were defined as those self-reported having been diagnosed or treated for osteoarthritis.

Controls were defined as those self-reporting as having not been diagnosed or treated for osteoarthritis.

The cohort included 253,880 cases and 1,286,245 controls.

Independent cohorts 2—OAI and JoCo:

*Cases*. Knee osteoarthritis was evaluated with fixed-flexion posteroanterior radiographs for OAI samples and for JoCo participants, weight-bearing anteroposterior extended radiographs were taken during initial recruitments and fixed-flexion posteroanterior radiographs were taken during follow-up. Cases were those with definitive knee osteoarthritis, defined as radiographic evidence of the presence of definite osteophytes and possible joint space narrowing (Kellgren-Lawrence grade ≥ 2) or total joint replacement in one or both knees. Controls were those having no or doubtful evidence of OA (Kellgren-Lawrence grade = 0 or 1) in both knees at all available time points.

These definitions were previously used by Yau et al.[29]. In the current study, there were 2672 cases (2014 from OAI and 658 from JoCo) and 1776 controls (953 from OAI and 823 from JoCo).

**Statistical analysis**. *GWAS analysis*: In the discovery stage, the BGENIE (https://jmarchini.org/bgenie/) was used as the main GWAS software, which was designed for analysing the UK Biobank genetic datasets. Routine QC steps included removal of SNPs with INFO scores <0.1, SNPs with minor allele frequency <0.5% or SNPs that failed Hardy–Weinberg tests ($P < 10^{-6}$). SNPs on the X and Y chromosomes and mitochondrial SNPs were also removed. We further removed data from individuals whose ancestry was not white British based on the principal component analysis, those who were related to at least one other participant in the cohort (a cutoff value of 0.025 in the generation of the genetic relationship matrix), and those who failed QC. Association tests based on the linear association were performed using BGENIE adjusting for age, sex, BMI, nine population principal components, genotyping arrays and assessment centres. A $\chi^2$ test was used to test for gender differences between cases and controls. Age and BMI were compared using independent t-test in IBM SPSS 22 (IBM Corporation, New York). A $P$ value $< 5 \times 10^{-8}$ was considered to indicate a genome-wide association. Independent SNPs were

defined as those that were not correlated ($r^2 < 0.6$) with any other associated SNP. GCTA (https://cnsgenomics.com/software/gcta/#Overview) was used to calculate the narrow-sense heritability using a genomic relationship matrix calculated from genotyped autosomal SNPs.

In the supporting stage (using independent cohorts), details of the identified genome-wide positive and independent SNPs associated with knee pain from the discovery stage were sent to 23andMe Inc and the combined OAI and JoCo cohorts. The genome-wide positive and independent SNPs were defined as those with $P$ value $< 5 \times 10^{-8}$ and with linkage disequilibrium value $r^2 < 0.6$. The 23andMe and the combined OAI and JoCo cohorts then extracted the summary statistics of these SNPs from their GWAS results, correspondingly.

23andMe performed GWAS on self-reported osteoarthritis in any joint using the logistic regression method assuming an additive genetic model for allelic effects adjusting for age, sex, five principal components and foue DNA chip platforms. Participants were restricted to a set of individuals who had >97% European ancestry, as determined through an analysis of local ancestry. A maximal set of unrelated individuals was chosen for the GWAS analysis using a segmental identity-by-descent estimation algorithm. Further details can be found in in the supplementary file of a previous publication[51].

The OAI and JoCo performed GWAS on radiographic knee osteoarthritis using logistic regression assuming an additive genetic model for allelic effects adjusting for age, sex, study site and principal components. Summary statistics from both cohorts were then combined in a meta-analysis. Only participants of Caucasian origin were included in the GWAS study. Standard procedures were used to remove data from non-Caucasian individuals and related individuals. Further details can be found in ref. [29].

*GWAS-associated analysis*: The FUMA web application was used as the main annotation tool, and a Manhattan plot and a Q–Q plot were also generated by this[53]. LocusZoom (http://locuszoom.org/) was used to provide regional visualisation.

*FUMA mainly provides three types of analysis*: the gene analysis, the gene-set analysis and the tissue expression analysis. In gene analysis, summary statistics of SNPs were aggregated to the level of whole genes to test the associations between genes with the phenotype. In gene-set analysis, groups of genes sharing certain biological, functional or other characteristics were tested together to provide insight into the involvement of specific biological pathways or cellular functions in the genetic aetiology of a phenotype. The tissue expression analysis was based on GTEx (https://www.gtexportal.org/home/), which is integrated into FUMA. Average gene expression per tissue type was used as gene covariate to test for relationships between gene expression in a specific tissue type and genetic associations with knee pain.

To identify genetic correlations between knee pain and all other 234 complex traits, we used linkage disequilibrium score regression through LD Hub v1.9.0 (available at http://ldsc.broadinstitute.org/ldhub/)[54]. The LD Hub estimates the bivariate genetic correlations of a phenotype with 234 traits using individual SNP allele effect sizes and the average linkage disequilibrium in a region. Those with $P$ values less than $2.1 \times 10^{-4}$ (0.05/234) were considered significant surviving Bonferroni correction for multiple testing.

**Reporting summary**. Further information on research design is available in the Nature Research Reporting Summary linked to this article.

## Data availability

The summary statistics of the UK Biobank results on knee pain can be accessed through https://figshare.com/articles/kneepaingwas/9611198. Data from 23andMe were obtained under a data transfer agreement. Further information about obtaining access to the 23andMe Inc. summary statistics is available from: https://research.23andme.com/collaborate/. Any other data relevant to the study that are not included in the article or its supplementary materials are available from the authors upon reasonable request.

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

## Acknowledgements

We would like to thank all participants of the UK Biobank, 23andMe and the OAI and JoCo cohorts who have provided necessary genetic and phenotypic information. The current study was conducted under approved UK Biobank data application number 4844. This work was supported by the STRADL project (Wellcome Trust, grant number: 104036/Z/14/Z), the DOLORisk project (Horizon 2020, grant number 633491) and the GCRF academic exchange visits programme by the University of Dundee. The Osteoarthritis Initiative (OAI) was public-private partnership comprised of five contracts (N01-AR-2–2258; N01-AR-2–2259; N01-AR-2–2260; N01-AR-2–2261; N01-AR-2–2262) funded by the NIH. The Johnston County Osteoarthritis Study (JoCo) was supported in part by S043, S1734, & S3486 from the CDC/Association of Schools of Public Health; 5-P60-AR30701 & 5-P60-AR49465–03 from NIAMS/NIH; genotyping was supported by Algynomics Inc. Additional support was obtained from NIH grant P30-DK072488.

## Author contributions

W.M. organised project, drafted the paper and contributed to the analysis. M.A. performed the main UK Biobank GWAS analysis. C.P., B.M. and M.Y. provided essential comments. K.R., J.J., B.M., R.J. and M.Y. provided the OAI and JoCo GWAS summary statistics on knee osteoarthritis. J.S. and A.A. provided results in the 23andMe cohort. A.M. and B.S. organised the project and provided comments.

## Additional information

**Competing interests:** J.S., A.A. and members of the 23andMe Research Team are employees of 23andMe, Inc., and hold stock or stock options in 23andMe. Other authors declare no competing interests.

## The 23andMe Research Team

Michelle Agee[3], Babak Alipanahi[3], Robert K. Bell[3], Katarzyna Bryc[3], Sarah L. Elson[3], Pierre Fontanillas[3], Nicholas A. Furlotte[3], Barry Hicks[3], David A. Hinds[3], Karen E. Huber[3], Ethan M. Jewett[3], Yunxuan Jiang[3], Aaron Kleinman[3], Keng-Han Lin[3], Nadia K. Litterman[3], Jennifer C. McCreight[3], Matthew H. McIntyre[3], Kimberly F. McManus[3], Joanna L. Mountain[3], Elizabeth S. Noblin[3], Carrie A. M Northover[3], Steven J. Pitts[3], G. David Poznik[3], J. Fah Sathirapongsasuti[3], Janie F. Shelton[3], Suyash Shringarpure[3], Chao Tian[3], Joyce Y. Tung[3], Vladimir Vacic[3], Xin Wang[3] & Catherine H. Wilson[3]

