## [Peer Review File · Communications Biology]

Reviewers' comments:

Reviewer #1 (Remarks to the Author):

This article describes a large genetic study seeking to identify genetic susceptibility loci associated with knee pain, with initial discovery being performed in the UK biobank and replication in 23andMe, OAI, and Johnson County OA studies.

It is an interesting study and a question that has not yet been answered in a broad GWAS manner, although several smaller studies have evaluated changes in smaller patient cohorts.

The findings of this study center around the identification of 2 loci within GDF5, which were confirmed in subsequent OAI and JoCo cohorts, and two within COL27A1, which were not able to be confirmed in subsequent cohorts.

I have a few specific problems with this study, which I outline below:

First, within the introduction, the authors point out that knee OA heritability is "as high as 0.62"; they should update this to include newer twin studies, which report a much lower genetic association. For example, Magnusson et al. 2017 suggested a concordance rate for severe knee OA requiring joint replacement of ~45%.

The authors go to lengths within the discussion section to point out that knee OA and knee pain are not entirely coincident; however, their 23andMe cohort selected cases as those being diagnosed or treated for OA, not joint pain specifically. More concerning in this regard, the question asked within the 23andMe cohort asked nothing about the location of OA. I will grant the authors that most cases of symptomatic, diagnosed OA in the US are knee, but a significant portion of patients will also have hip and/or hand OA, which are likely to have a different genetic makeup. Given this discrepancy, I feel that the 23andMe cohort should not be included in this analysis. Both the JoCo and OAI cohorts have much better-defined, knee-specific pain criteria, albeit with smaller sample sizes. I would guess that removing the 23andMe replication cohort would reduce the significance of the meta-analysis p-value somewhat.

Alternatively, the authors might go back and analyze the UK biobank for signals within GDF5 and COL27A1 which correlate with joint pain at sites other than knees; i.e. hip and/or hand, which might indicate that these genetic associations are correlated with joint pain more generally rather than knee pain specifically.

Although the introduction / methods / results sections are well-written, there are several grammatical errors within the discussion section that need to be corrected (i.e. line 330...COL27A1 gene might play a role in the knee area [sentence fragment], line 338 "...attachment to the patellar..." [presumably meant "patella"]).

The authors correctly indicate within the discussion section that genes identified and replicated in their study of knee pain are those related to knee OA. They wonder whether other causes of knee pain might be coloring their results; however, I would caution that early knee pain cases may simply be early knee OA which does not yet have radiographic incident findings. For example, Thorstensson et al. 2009: 86% of patients initially presenting with knee pain go on to develop incident radiographic OA within the subsequent 12 years.

Reviewer #2 (Remarks to the Author):

The authors reported two loci associated with self-reported knee pain without radiographical diagnosis at genome-wide significant level in a large UK Biobank cohort. They replicated the result in one independent cohort with self-reported osteoarthritis with unknown pain status (23&me) and two cohorts with radiographically diagnosed knee osteoarthritis (OAI and JoCo) with unknown pain status. The population is very heterogeneous, but it is compensated by large sample size. The result is statistically significant but with very small effect size.

I have the following comments:

-Replication cohorts are not true replications. The 23&me cohort are self-reported people with diagnosed osteoarthritis of unknown location. It can be the low back, neck, wrist, elbow, shoulder, hip, knee or feet. In fact, they may not be painful at all since 23&me did not include such a question. It's an association study with "osteoarthritis". It is misleading to call such a cohort as "knee pain" replication. The OAI and JoCo cohort are also not really replication, since radiographically diagnosed knee osteoarthritis can be not painful. It is an association study with "knee osteoarthritis", and not "knee pain". As authors rightly pointed out, only 50% of knee osteoarthritis patients with radiographic evidence had knee pain symptoms. Meta-analysis is therefore inappropriate because phenotypes are not the same.

-I would recommend the authors avoid using the wording of "replication" and "meta-analysis" throughout the manuscript. I would treat the "replication cohorts" as independent cohorts with somewhat different phenotype and discuss similar findings very carefully in the discussion section. It may be appropriate to remove the replication cohort entirely and publish them in a separate paper as osteoarthritis association.

-Table 2: Effect size was reported with Beta, which is usually calculated from linear regression. However, no linear regression was used and effect size was reported with odds ratio in the result section. Please clarify.

-This study replicated multiple previous studies that rs143384 in GDF5 as a risk SNP for osteoarthritis (based on 23&me/OAI/JoCo cohorts). What is new is the association near COL27A1. Another collagen gene, COL11A2, has been recently found to be associated with pain sensitivity in knee osteoarthritis (<https://doi.org/10.1177/1744806917724259>). None of these two genes are in the highest-associated pathways, but they may suggest a role of collagen gene in development of knee pain.

Reviewer #3 (Remarks to the Author):

The authors reported a genome-wide association study of knee pain in the UK Biobank samples. They tried to replicate the top significant SNPs in to replication cohorts of 23andme and OAI & JoCo cohorts. They reported that they found two GW-significant loci. They also estimated the contribution of common genetic variants on the phenotypic variance (i.e. chip heritability). They found that 8% of the variation in knee pains is due to common genetic variants. Further, they performed LD score regression to test for the genetic correlation between knee pain and 234 phenotypes within the UK Biobank and found some significant negative correlation with traits, such as educational attainment.

Main comments

This is a standard GWAS analysis using publicly available data. I do, however, have a major concern about the difference in the phenotypes between the discovery and two replication cohorts. In the UK Biobank samples, knee pain is defined 'knee pain in the last 72 month interfering with usual activity'. However, the phenotypes in the replication cohorts are 'self-reported having been diagnosed or

treated for osteoarthritis' and 'Knee osteoarthritis' in the 23andme samples and OAI & JoCo, respectively. While the authors argued in the discussion that the phenotypes may be similar, but I am not convinced since knee pain can be attributed to many causes, not only osteoarthritis.

Additional comments

Introduction. What is the heritability of knee pain from twin and family studies, so we have a benchmark to compare the 8% chip heritability that they estimated using GCTA.

Page 8. It is not clear what 'standard Frequentist association' is?

Page 8-9. The model (and covariates) used in the discovery and replication cohorts are different. For example, 5 PCs in UKB, 9 PCs in 23andME and unknown number of PCs in OAI&JoCo cohorts.

Page 8. "Linkage disequilibrium value $r^2 < 0.6$." is still highly correlated to be called 'independent SNPs'

Page 9. The beginning of paragraph 2 is started with "..OAI and JoCo performed GWAS on radiographic", is it a typo?

Page 10. Were sample overlaps checked in LD score regression analyses?

Reviewer 1

1. First, within the introduction, the authors point out that knee OA heritability is "as high as 0.62"; they should update this to include newer twin studies, which report a much lower genetic association. For example, Magnusson et al. 2017 suggested a concordance rate for severe knee OA requiring joint replacement of ~45%.

Reply: Thank you. We have added this useful paper as a reference.
(page 3, line 57)

2. The authors go to lengths within the discussion section to point out that knee OA and knee pain are not entirely coincident; however, their 23andMe cohort selected cases as those being diagnosed or treated for OA, not joint pain specifically. More concerning in this regard, the question asked within the 23andMe cohort asked nothing about the location of OA. I will grant the authors that most cases of symptomatic, diagnosed OA in the US are knee, but a significant portion of patients will also have hip and/or hand OA, which are likely to have a different genetic makeup. Given this discrepancy, I feel that the 23andMe cohort should not be included in this analysis. Both the JoCo and OAI cohorts have much better-defined, knee-specific pain criteria, albeit with smaller sample sizes. I would guess that removing the 23andMe replication cohort would reduce the significance of the meta-analysis p-value somewhat.

Reply: Indeed, as authors, we did anticipate that this concern would be raised. At the moment, to the best of our knowledge, there is no GWAS

summary statistics dataset available on knee pain yet. This means that there is no way to find a good GWAS replication cohort on which to test the results from our discovery study (UK Biobank). To compensate for this, in the initial submitted manuscript, we chose to use knee osteoarthritis cohorts and osteoarthritis cohorts to provide supporting evidence for the validity of our findings. We are aware the limitations of each cohort as we have described them in the manuscript.

The reviewer 1 is right about removing the 23andMe replication cohort would reduce the significance of the meta-analysis p-value.

To address this, Reviewer 2 suggested a change of language, avoiding use of the words "replication" and "meta-analysis" throughout the manuscript (use "independent" instead). This approach would maintain the main structure of the manuscript, presenting the findings from the discovery GWAS while still fulfilling the need for validation studies as well as is possible with currently available cohorts.

So, we have adopted the 'wording' suggestion made by Reviewer 2 (Consulted and agreed with the Chief Editor), and changed the language throughout the paper. We hope that the reviewer finds this to be an acceptable solution.

Alternatively, the authors might go back and analyze the UK biobank for signals within GDF5 and COL27A1 which correlate with joint pain at sites other than knees; i.e. hip and/or hand, which might indicate that these genetic associations are correlated with joint pain more generally rather than knee pain specifically.

Reply: This is a good suggestion to study the pain phenotypes at different joints. Currently we only have approval to access data relating to the pain phenotypes in the UK Biobank and we do not have permission to use the OA phenotypes. But we can collaborate with those who have the permission to work on this in the future.

3. Although the introduction / methods / results sections are well-written, there are several grammatical errors within the discussion section that need to be corrected (i.e. line 330...COL27A1 gene might play a role in the knee area [sentence fragment], line 338 "...attachment to the patellar..." [presumably meant "patella"]).

Reply: Thank you. The sentences have been modified. (page 14, line 323-324; page 15, line 332)

4. The authors correctly indicate within the discussion section that genes identified and replicated in their study of knee pain are those related to knee OA. They wonder whether other causes of knee pain might be coloring their results; however, I would caution that early knee pain cases may simply be early knee OA which does not yet have radiographic incident findings. For example, Thorstensson et al. 2009: 86% of patients initially presenting with knee pain go on to develop incident radiographic OA within the subsequent 12 years.

Reply: Good suggestion. We did discuss this carefully in the manuscript. We have cited the paper to strength the discussion. (page 15, line 345-346)

Reviewer 2

1. Replication cohorts are not true replications. The 23&me cohort are self-reported people with diagnosed osteoarthritis of unknown location. It can be the low back, neck, wrist, elbow, shoulder, hip, knee or feet. In fact, they may not be painful at all since 23&me did not include such a question. It's an association study with "osteoarthritis". It is misleading to call such a cohort as "knee pain" replication. The OAI and JoCo cohort are also not really replication, since radiographically diagnosed knee osteoarthritis can be not painful. It is an association study with "knee osteoarthritis", and not "knee pain". As authors rightly pointed out, only 50% of knee osteoarthritis patients with radiographic evidence had knee pain symptoms. Meta-analysis is therefore inappropriate because phenotypes are not the same.

I would recommend the authors avoid using the wording of "replication" and "meta-analysis" throughout the manuscript. I would treat the "replication cohorts" as independent cohorts with somewhat different phenotype and discuss similar findings very carefully in the discussion section. It may be appropriate to remove the replication cohort entirely and publish them in a separate paper as osteoarthritis association.

Reply: Good suggestion. Indeed, as authors, we did anticipate that this concern would be raised. At the moment, to the best of our knowledge, there is no GWAS summary statistics dataset available on knee pain yet. This means that there is no way to find a good GWAS replication cohort on which to test the results from our discovery study (UK Biobank). To compensate for this, in the initial submitted manuscript, we chose to use knee osteoarthritis

cohorts and osteoarthritis cohorts to provide supporting evidence for the validity of our findings. We are aware the limitations of each cohort as we have described them in the manuscript. We are aware that any new GWAS will normally require a replication or validation study in order to be suitable for publication.

We chose to replace the words "replication" with "independent" or "supporting" as necessary **throughout the manuscript**. We have removed the meta-analysis from the manuscript.

2. Table 2: Effect size was reported with Beta, which is usually calculated from linear regression. However, no linear regression was used and effect size was reported with odds ratio in the result section. Please clarify.

Reply: The software we use is BGENIE (designed for the UK Biobank datasets), which performs a linear association test between SNP/phenotype pairs in the provided data. So betas are the direct outputs from the software. As most of the readers will be more familiar with odds ratios, we have replaced betas with odds ratios in the Table 2.

3. This study replicated multiple previous studies that rs143384 in GDF5 as a risk SNP for osteoarthritis (based on 23&me/OAI/JoCo cohorts). What is new is the association near COL27A1. Another collage gene, COL11A2, has been recently found to be associated with pain sensitivity in knee osteoarthritis. (<https://doi.org/10.1177/1744806917724259>). None of these two genes are in the highest-associated pathways, but they may suggest a role of collagen gene in development of knee pain.

Reply: Good suggestion. We have cited this paper in the discussion part. (page 14, line 333-334)

Reviewer 3

1. This is a standard GWAS analysis using publicly available data. I do, however, have a major concern about the difference in the phenotypes between the discovery and two replication cohorts. In the UK Biobank samples, knee pain is defined 'knee pain in the last 72 month interfering with usual activity'. However, the phenotypes in the replication cohorts are 'self-reported having been diagnosed or treated for osteoarthritis' and 'Knee osteoarthritis' in the 23andme samples and OAI & JoCo, respectively. While the authors argued in the discussion that the phenotypes may be similar, but I am not convinced since knee pain can be attributed to many causes, not only osteoarthritis.

Reply: Indeed, as authors, we did anticipate that this concern would be raised. At the moment, to the best of our knowledge, there is no GWAS summary statistics dataset available on knee pain yet. This means that there is no way to find a good GWAS replication cohort on which to test the results from our discovery study (UK Biobank). To compensate for this, in the initial submitted manuscript, we chose to use knee osteoarthritis cohorts and osteoarthritis cohorts to provide supporting evidence for the validity of our findings. We are aware the limitations of each cohort as we have described them in the manuscript. We are aware that any new GWAS will normally require a replication or validation study in order to be suitable for publication.

We agree that the cause of knee pain can be heterogeneous as many pain phenotypes such as back pain. We think that our genetic study can provide useful information and evidence to understand the genetic mechanisms of knee pain.

2. Introduction. What is the heritability of knee pain from twin and family studies, so we have a benchmark to compare the 8% chip heritability that they estimated using GCTA.

Reply: There is no published genetic study of knee pain (as we have mentioned in the paper), so most of the heritability information we can use is from results of knee osteoarthritis studies. We have cited a twin study in the revision version, which suggested that 45% of the respective variation for severe knee osteoarthritis requiring joint replacement can be explained by genetic factors on knee osteoarthritis.

3. Page 8. It is not clear what 'standard Frequentist association' is?

Reply: Thank you. We have replaced the 'standard Frequentist association' with 'linear association'. (page 8, line 167-168)

4. Page 8-9. The model (and covariates) used in the discovery and replication cohorts are different. For example, 5 PCs in UKB, 9 PCs in 23andME and unknown number of PCs in OAI&JoCo cohorts

Reply: We used 9 PCs for the UK Biobank, 5 PCs for 23andMe. The OAI & JoCo summary statistics results were the meta-analysis results of both cohorts. So there is no PCs information for the combined OAI &

JoCo summary statistics. In fact, we have removed the whole meta-analysis part from the manuscript as suggested by Reviewer 2.

However, we do believe that the number of PCs might not be impactful in a meta-analysis. The number of PCs is about what is sufficient to control inflation within each study, it isn't necessary for them to be consistent across studies. It is also unrealistic to require different participating cohorts to use the same number of PCs. For example, Zeggini et al. Nat Genet. 2008 May; 40(5): 638–645. It included many cohorts with GWAS on diabetes using different numbers of PCs.

5. Page 8. “Linkage disequilibrium value $r^2 < 0.6$.” is still highly correlated to be called ‘independent SNPs’

Reply: The LD value of $R^2 < 0.6$ as independent SNPs is defined by FUMA as default. Researchers can modify the value as they wish. A higher value will decrease the number of independent SNPs and a low value will increase the number of independent SNPs.

6. Page 9. The beginning of paragraph 2 is started with “..OAI and JoCo performed GWAS on radiographic”, is it a typo?

Reply: Thank you. We have modified this sentence. (page 9, line 193)

7. Page 10. Were sample overlaps checked in LD score regression analyses?

Reply: The LD score regression estimates are in general robust to sample overlaps. The amount of overlap can be assessed by examining the genetic covariance intercept in the LDHub outputs. If the

cross trait intercept is close to zero (the intercept is less than one standard error away from zero), it indicates there is no overlap

For the 3 traits that genetically correlated with knee pain, those numbers were very close to zero.

Trait 1	Trait 2	Genetic covariance Intercept	Intercept SE
kneepain	Years of schooling 2016	0.0035	0.0059
kneepain	Age of having first baby	-0.0007	0.0056
kneepain	College completion	-0.0024	0.0057

Thank you all again.

Yours sincerely

The corresponding author and coauthors

Reviewers' comments:

Reviewer #1 (Remarks to the Author):

The authors have responded to my comments adequately. I still worry about the strength of their confirmation findings with the 23andme cohort; however, they have duly noted the limitations of using this dataset in their manuscript.

Reviewer #2 (Remarks to the Author):

Abstract:

"These findings were subsequently supported in independent cohorts."- This needs to be rephrased. It should be stated to the effect of "These findings were supported in two independent cohorts with self-reported osteoarthritis/knee osteoarthritis without pain information.". One or two more sentences should be added to discuss the implications similarity of findings from two different phenotypes. Something like this can be added "although not true replications, these independent cohorts suggest possible overlapping risk alleles among knee pain, general osteoarthritis and knee osteoarthritis"

This additional sentence is not a major finding and I believe it shouldn't be added to the abstract " Based on the LD Hub, knee pain was genetically correlated with phenotypes of Years of schooling, College completion and Age of having first baby."

Methods:

Regarding association analysis (major concern):

This is a case control study, so I assume it was performed with logistic regression and not linear regression. I've never used BGENIE, but I went into their website/original paper and it seems to only be able to compute with linear regression and not logistic regression (please correct me if I'm wrong). If the output was beta, then it was beta. It should not be changed into odds ratio as output. If the authors are doing logistic regression (case-control study), they should have used SNPTTEST (listed within the same website) or programs with similar functions to perform the analysis.

Results:

Line 248: Phenotype needs to be clearly stated " In the independent cohort 1 "with self-reported osteoarthritis", the P value of rs143384 was 2.44×10^{-9} in the 23andMe cohort. "In the independent cohorts 2 with radiographically diagnosed knee osteoarthritis", the P value of rs143384 was 0.01 in the combined OAI and JoCo cohorts.

This sentence should be deleted because it's not appropriate to combine statistics with different phenotypes: "The P values of rs2808772 were 4.43×10^{-5} in the 23andMe and 0.36 in the combined OAI and JoCo cohorts. "

Reviewer #3 (Remarks to the Author):

The authors have responded to my comments and also the other reviewers' comments regarding the difference between the phenotype of the discovery and replication cohorts.

However, their response was only by changing the wording from 'replication' to 'independent' cohorts. This change of the wording does not change the overall structure of the paper and it may in fact confuse the readers. The readers may think that the independent cohort is just a different word for

the replication cohort. Therefore, I have still reservation regarding the inclusion of the 23andme cohort in this manuscript.

Reviewer 1

The authors have responded to my comments adequately. I still worry about the strength of their confirmation findings with the 23andme cohort; however, they have duly noted the limitations of using this dataset in their manuscript.

Reply: Many thanks. The 23andme cohort is by far the largest available cohort reporting data on osteoarthritis, containing over 1 million subjects. The genetic results for osteoarthritis have not previously been published from this cohort, although we recognize the limitations of this self-reported phenotype, as noted in the discussion .

Reviewer 2

Abstract:

"These findings were subsequently supported in independent cohorts."- This needs to be rephrased. It should be stated to the effect of "These findings were supported in two independent cohorts with self-reported osteoarthritis/knee osteoarthritis without pain information."

Reply: Thank you. we have modified this sentence as suggested.

One or two more sentences should be added to discuss the implications similarity of findings from two different phenotypes. Something like this can be added "although not true replications, these independent cohorts suggest possible overlapping risk alleles among knee pain, general osteoarthritis and knee osteoarthritis."

Reply: Good suggestion. We have added this sentence into the discussion.

Page 11, line 253-255.

This additional sentence is not a major finding and I believe it shouldn't be added to the abstract " Based on the LD Hub, knee pain was genetically correlated with phenotypes of Years of schooling, College completion and Age of having first baby."

Reply: We have deleted this sentence in the abstract.

Methods:

Regarding association analysis (major concern):

This is a case control study, so I assume it was performed with logistic regression and not linear regression. I've never used BGENIE, but I went into their website/original paper and it seems to only be able to compute with linear regression and not logistic regression (please correct me if I'm wrong). If the output was beta, then it was beta. It should not be changed into odds ratio as output. If the authors are doing logistic regression (case-control study), they should have used SNPTTEST (listed within the same website) or programs with similar functions to perform the analysis.

Reply: Indeed, it is a case-control study. Many software nowadays adapt linear regression for binary phenotypes because it can efficiently and robustly account for population stratification and relatedness through inclusion of random effects for a genetic relationship matrix.

(<https://www.ncbi.nlm.nih.gov/pmc/articles/PMC5237383/>;

<https://www.nature.com/articles/s41588-018-0144-6>). Software examples include GCTA, BOLT-LMM, etc.

One advantage about using the BGENIE is that it directly uses the provided UK Biobank files as input and avoids repeated decompression and conversion of these files during analysis. A previous example of using BGENIE to perform a GWAS on a binary phenotype (self-reported neuroticism) is here. (<https://www.nature.com/articles/s41467-018-04362-x>).

It is advised that although the SNPTEST is a good software for GWAS, the computer time required to run SNPTEST does not scale well to biobank-sized datasets.

The reason for using odds ratio (OR) is that most readers are more familiar with OR instead of 'beta'. However, we do agree with the reviewer 2 that we should use 'beta' in this case, so we have now used 'beta' in the manuscript.

Results

Line 248: Phenotype needs to be clearly stated " In the independent cohort 1 "with self-reported osteoarthritis", the P value of rs143384 was 2.44×10^{-9} in the 23andMe cohort. "In the independent cohorts 2 with radiographically diagnosed knee osteoarthritis", the P value of rs143384 was 0.01 in the combined OAI and JoCo cohorts.

Reply: We have modified the sentence as suggested. Page 5, line 111-113.

This sentence should be deleted because it's not appropriate to combine statistics with different phenotypes: "The P values of rs2808772 were 4.43×10^{-5} in the 23andMe and 0.36 in the combined OAI and JoCo cohorts. "

Reply: We did not combine statistics here. The OAI and JoCo cohorts used the same phenotype definition. The P values of rs2808772 were generated independently in 23andMe and in the combined OAI and JoCo cohorts. To avoid misunderstanding, we added 'self-reported osteoarthritis' after 23andMe and 'radiographic knee osteoarthritis' after OAI and JoCo cohorts.

Reviewer 3

The authors have responded to my comments and also the other reviewers' comments regarding the difference between the phenotype of the discovery and replication cohorts.

However, their response was only by changing the wording from 'replication' to 'independent' cohorts. This change of the wording does not change the overall structure of the paper and it may in fact confuse the readers. The readers may think that the independent cohort is just a different word for the replication cohort. Therefore, I have still reservation regarding the inclusion of the 23andme cohort in this manuscript.

Reply: We acknowledge this important point raised by the reviewer. In our revision, we have removed the term 'replication,' and inserted into the Discussion the explicit statement "Although not true replications, these independent cohorts suggest possible overlapping risk alleles among knee pain, general osteoarthritis and knee osteoarthritis." (page 11, lines 253-255). To further pre-empt any confusion from readers, we have also reminded readers in the Results section (page 5, lines 111-117) that independent

cohorts 1 and 2 have different case definitions: “In the independent cohort 1 (23andMe, self-reported osteoarthritis), the *P* value ... in the 23andMe cohort and in the independent cohorts 2 (OAI and JoCo radiographic knee osteoarthritis) ,...” We hope that the reviewer will agree that these changes will minimize confusion to the readers.

Thank you all again.

Yours sincerely

The corresponding author and coauthors

REVIEWERS' COMMENTS:

Reviewer #2 (Remarks to the Author):

The authors have appropriately addressed my concerns.